# Mixotrophy in Marine Microalgae to Enhance Their Bioactivity

**DOI:** 10.3390/microorganisms13020338

**Published:** 2025-02-05

**Authors:** Gabriella Licata, Christian Galasso, Fortunato Palma Esposito, Antonio Palumbo Piccionello, Valeria Villanova

**Affiliations:** 1Department of Biological, Chemical and Pharmaceutical Sciences and Technologies (STEBICEF), University of Palermo, 90128 Palermo, Italy; gabriella.licata02@you.unipa.it (G.L.); antonio.palumbopiccionello@unipa.it (A.P.P.); 2Department of Ecosustainable Marine Biotechnology, Stazione Zoologica Anton Dohrn, Calabria Marine Centre, C. da Torre Spaccata, 87071 Amendolara, Italy; christian.galasso@szn.it; 3Department of Ecosustainable Marine Biotechnology, Stazione Zoologica Anton Dohrn, Via Ammiraglio Acton 55, 80133 Naples, Italy; fortunato.palmaesposito@szn.it

**Keywords:** microalgae, photoheterotrophy, biological activity, cytotoxicity, antibacterial activity, anticancer activity, MS-HPLC, *Phaeodactylum tricornutum*, *Chlorella* sp., *Nannochloropsis granulata*

## Abstract

Photosynthetic microorganisms, such as microalgae, are remarkable for their ability to harness sunlight, fix carbon dioxide, and produce a variety of bioactive compounds. These organisms are pivotal in climate mitigation strategies as they can absorb carbon dioxide while generating valuable biomolecules. Among the diverse cultivation approaches, mixotrophic growth combines light energy with both inorganic and organic carbon sources, offering a unique strategy to enhance biomass production and metabolic diversity in microalgae. Here, microalgal species such as *Nannochloropsis granulata*, *Phaeodactylum tricornutum*, and *Chlorella* sp. were investigated for their potential applications under different cultivation methods, including phototrophy and mixotrophy. Mixotrophic conditions significantly improved biomass production across all tested species. Among these, *Phaeodactylum tricornutum*, a marine diatom, emerged as a promising candidate for bioactive compound production, exhibiting higher antiproliferative activity against human melanoma cells and antibacterial effects against *Staphylococcus aureus*. Importantly, *Chlorella* sp. was also found to possess antibacterial activity against *Staphylococcus aureus*, broadening its potential applications. Additionally, metabolomics analysis was performed on *Chlorella* sp. and *Phaeodactylum tricornutum* to identify the compounds responsible for the observed bioactivity. This study highlights the value of mixotrophic cultivation in enhancing the productivity and bioactivity of microalgae, positioning them as versatile organisms for sustainable biotechnological applications.

## 1. Introduction

Photosynthetic microorganisms, particularly microalgae, play an essential role in global ecosystems due to their ability to capture carbon dioxide and produce oxygen via photosynthesis, contributing approximately 50% of the oxygen in the Earth’s atmosphere [1]. These microorganisms are not only crucial in mitigating climate change but also serve as a promising resource for sustainable biotechnology. Microalgae species are characterized by rapid growth, producing substantial biomass enriched with high-value biomolecules such as carotenoids, polyunsaturated fatty acids, vitamins, lipids, and proteins [2]. These compounds have diverse applications across industrial sectors, from biofuels to pharmaceuticals, making microalgae key players in the transition to a sustainable economy [3].

The rise in antibiotic-resistant bacteria underscores the urgent need to discover new bioactive compounds that are safe, effective, and cost-efficient. Microalgae are known to produce a variety of antibacterial agents with identified molecules, including phenolic compounds, aplysiatoxin, phlorotannins, peptides, terpenes, polysaccharides, polyacetylenes, sterols, alkaloids, aromatic organic acids, shikimic acid, polyketides, hydroquinones, and fatty acids [4]. In addition to their antibacterial potential, microalgae also hold promise in anticancer drug development. Chemical diversity of microalgal compounds can trigger complex intracellular mechanisms of action, such as the selective inhibition of cancer proliferation and cell death [5,6]. Among the diverse microalgal species, *Phaeodactylum tricornutum* (*P. tricornutum*), *Chlorella* sp., and *Nannochloropsis granulata* (*N. granulata*) have attracted significant research interest due to their biotechnological potential [7,8]. To maximize their potential, researchers have explored different cultivation modes, including phototrophy and mixotrophy. Among these, mixotrophy has proven particularly effective, as it combines light with inorganic and organic carbon sources to boost both biomass production and metabolite diversity [9,10].

*P. tricornutum* is a marine diatom known for its remarkable metabolic versatility and ability to produce bioactive compounds, including polyunsaturated fatty acids like eicosapentaenoic acid (EPA) and pigments such as fucoxanthin [11]. Its genome has been sequenced, providing insights into its metabolic pathways and enhancing its utility for genetic engineering [12]. Moreover, this diatom is considered a promising source of several bioactive molecules. For example, *P. tricornutum* has been shown to produce antimicrobial peptides with significant antibacterial effects, further highlighting its versatility as a source of bioactive compounds [13].

*Chlorella*, a genus of green microalgae, is well known for its rapid growth and high adaptability to different environmental conditions. It is widely utilized for its nutritional properties, producing proteins, lipids, and antioxidants, and has been explored for biofuel production, wastewater treatment, and bioactive compound synthesis [7,14]. Previous studies have highlighted its antibacterial and anticancer properties, demonstrating its value in pharmaceutical and health-related applications [15,16].

*N. granulata*, a species within the genus *Nannochloropsis*, is a marine microalga known for its high lipid content, particularly EPA, a valuable omega-3 fatty acid. This genus is frequently used in aquaculture as a nutritional supplement and is gaining attention for its potential in biofuel production and biotechnological applications [17,18,19]. Its robust growth under varied trophic conditions, coupled with its ability to produce bioactive metabolites, makes it an attractive candidate for industrial applications [10].

## 2. Materials and Methods

### 2.1. Microalgal Strains and Preculture Cultivation

*P. tricornutum*, *Chlorella* sp., and *N. granulata* were purchased from the Gothenburg University Marine Algal Culture Collection (GUMACC https://www.gu.se/en/marina-vetenskaper/about-us/algal-bank-gumacc, accessed on 1 December 2024). *P. tricornutum* was previously isolated from Plymouth, England [20]. *Chlorella* sp. was previously isolated from Barcarello beach, Palermo [7]. *N. granulata* was initially isolated by [21] from the Skagerrak, northeast Atlantic Ocean. The cultures were not axenic; however, 100 µg/L of ampicillin was incorporated at the start of the cultivation to regulate bacterial growth.

Precultures were maintained in 40 mL flasks at 22 °C, with continuous light of about 20 µmol photons m^−2^ s^−1^. The medium used in this study was GoldMedium (GM) provided by Aqualgae (https://aqualgae.com/portfolio/culture-media-for-microalgae/, accessed on 1 February 2023) (GM was prepared using two stock solutions: solution A (macronutrients) and solution B (a premix of trace elements and vitamins), which were added to artificial seawater prepared with salt solution I and salt solution II, following the method described by [22].

Solution A was prepared by adding 19 g of a macronutrient mix consisting of sodium nitrate (NaNO_3_) and sodium dihydrogen phosphate (NaH_2_PO_4_·2H_2_O); it was dissolved in 500 mL of milli-Q water in a 1 L glass bottle.

Solution B 0.25 g of the premix containing trace elements and vitamins was dissolved in 500 mL of Milli-Q water in a separate 1 L glass bottle.

Solution A, solution B, salt solution I, salt solution II, and milli-Q water were autoclaved separately at 121 °C for 21 min. After cooling to room temperature, the solutions were combined under aseptic conditions: 1 mL of solution A, 1 mL of solution B, and 10 mL each of salt solutions I and II were mixed. The final volume was adjusted to 1000 mL with autoclaved milli-Q water.

### 2.2. Growth Cultivation in Flasks

*P. tricornutum*, *Chlorella* sp., and *N. granulata* were cultured in 40 mL cell culture flasks with vented filter caps to allow gas exchange while maintaining sterile conditions. The cells were grown under constant low-intensity light of about 20 µmol photons m^−2^ s^−1^ at a temperature of 22 ° C and shaking at 100 rpm to ensure proper mixing and aeration throughout the experiment.

Four different culture conditions were tested as follows:-Phototrophy: growth in GM without an external carbon source (GM, pH ~6.7).-Mixotrophy: growth in GM supplemented with 4.6 g/L of glycerol (GM+GLY, pH ~6.9).-Phototrophy with bicarbonate: growth in GM supplemented with 1.26 g/L of bicarbonate (GM+BIC, pH ~8.2).

Mixotrophy with glycerol and bicarbonate: growth in GM supplemented with both 4.6 g/L of glycerol and 1.26 g/L of bicarbonate (GM+BIC+GLY pH ~8.3).

The pH was not actively controlled but rather buffered using bicarbonate in GM+BIC and GM+BIC+GLY conditions. This ensures that pH fluctuations are minimized during cultivation, as bicarbonate is known to act as a buffer to stabilize pH [23].

GLY was chosen as the organic carbon source because it has been previously shown to be highly effective in supporting the growth of *N. granulata *and *P. tricornutum* [10,24]. In addition to its effectiveness, glycerol is a low-cost carbon source, making it an economical choice for large-scale cultivation, which is critical for industrial applications [25]. BIC was selected as the inorganic carbon source, with its concentration optimized through mathematical simulations for *P. tricornutum* to biomass production [9], which is a cheaper and more suitable inorganic carbon alternative to the CO_2_ [26]. Moreover, BIC is commonly used to maintain favorable pH conditions in microbial cultures [23].

All conditions were prepared in triplicate.

Algal growth was monitored daily by measuring the optical density at 750 nm (OD_750_) using a spectrometer and 1 mL cuvettes (standard 10 mm light path).

Biomass yield was assessed by processing 2–3 mL of liquid culture after 10 days of cultivation. The samples were filtered through pre-weighed 0.2 µm pore-size filters, rinsed with sterile physiological solution, and dried in an oven at 100 °C for 24 h according to [27]. The dried filters were then weighed, and the biomass yield was calculated in grams per liter (g/L) using the following formula:


$$Dry\;weight\;(\frac{g}{L})=\frac{\left(\mathrm{Weight}\;\mathrm{of}\;\mathrm{dried}\;\mathrm{filter}\;\mathrm{with}\;\mathrm{biomass},g)-(initial\;weight\;of\;filter,g\right)}{\left(\mathrm{Volume}\;\mathrm{of}\;\mathrm{filtered}\;\mathrm{liquid}\;\mathrm{culture},L\right)}.$$


### 2.3. Growth Cultivation in the Multicultivator System

The Multicultivator MC 1000-OD (Photon System Instruments, Drásov, Czech Republic) was utilized to scale up the two best-performing conditions from the flask experiments: GM+BIC and GM+BIC+GLY in the three strains. The instrument consists of eight individual 85 mL culture tubes equipped with built-in LED lights and temperature control. Cultures of *P. tricornutum*, *Chlorella* sp., and *N. granulata* were inoculated with an initial OD_750_ of approximately 0.1 (5–10% inoculation rate), ensuring consistent starting conditions for all treatments. The temperature was maintained at 22 °C, with continuous aeration achieved using filtered air delivered to each tube at a flow rate of 1.5 L/min to ensure proper aeration and mixing. Illumination was provided with constant white light of 100 µmol photons m^−2^ s^−1^. Growth rates were monitored regularly by measuring OD_750_ using Agilent Cary 60 UV–Vis, Milan, Italy. At the end of the cultivation period, biomass was harvested, and final biomass yield was calculated using the same method as in the flask experiments.

#### Biomass Yield per Unit of Available Carbon

The biomass yield per unit of available carbon was calculated to evaluate the efficiency of carbon utilization in microalgal biomass production in the Multicultivator system (i.e., GM+BIC and GM+BIC+GLY conditions). This parameter was expressed as a percentage of grams of biomass produced per gram of available carbon in the culture medium $\left(\frac{g}{gC},\;\%\right)$. The calculation was performed using the following formula:


$$Biomass\;yield\;per\;unit\;of\;available\;carbon\;(\frac{g}{gC},\;\%)=\frac{\left(Dry\;weight,\frac{g}{L}\right)}{\left(Available\;carbon,\;\frac{gC}{L}\right)}*100.$$


The concentration of available carbon in GM+BIC (gC/L) was derived from the initial concentrations of BIC (1.26 g/L). The concentration of available carbon in GM+BIC+GLY (gC/L) was derived from the sum of the contributions from BIC (1.26 g/L) and glycerol (4.6 g/L). Using this total carbon concentration, the biomass yield for each strain and growth condition was calculated. This approach provides a more specific measure of the efficiency with which each microalgal strain utilized the supplied carbon sources under the tested cultivation conditions.

### 2.4. Metabolite Extraction

Freeze-dried biomasses from growth in the Multicultivator system were resuspended in 100% methanol and homogenized using a glass pestle. Methanol was selected as the extraction solvent due to its non-selective nature, enabling the extraction of a broad range of metabolites with varying polarities. The samples were agitated in darkness at room temperature for 60 min to ensure thorough extraction of intracellular metabolites.

Following extraction, the samples were centrifuged at 6000× *g* for 10 min at 4 °C to separate cellular debris from the supernatant.

The resulting supernatants were concentrated using SpeedVac SPD 130 DLX—Thermo Scientific, Zaventem, Belgium to obtain dried total extracts. For bioactivity assays, dried extracts were dissolved in dimethyl sulfoxide (DMSO) at a concentration of 40 mg/mL. For chemical analysis, dried extracts were dissolved in methanol at 0.1 mg/mL.

### 2.5. Antibacterial Activity Assays

To evaluate the antimicrobial potential of the algal extracts, samples (stock concentration 100 mg/mL) were placed into each well of a 96-well microtiter plate at an initial concentration of 2% (*v*/*v*) and serially diluted using MH medium. The Gram-positive *Staphylococcus aureus* ATCC 25923 (*S. aureus*) and the Gram-negative *Escherichia coli* ATCC 25922 (*E. coli*) were used as target strains. A single colony of a pathogen was used to inoculate 3 mL of liquid MH medium in a sterile bacteriological tube. After 6–8 h of incubation, growth was measured by monitoring the absorbance at 600 nm (OD_600_), and about 4 × 10^5^ CFU were dispensed in each well of the prepared plate. Plates were incubated at 37 °C for 24 h, and growth was assessed by measuring the absorbance at 600 nm by using an ELX800 Absorbance Microplate Reader (Biotek, Winoosky, VT, USA). DMSO was used as a negative control and to determine the effect of solvent on pathogen growth. Wells containing only growth medium represented a control to exclude medium contamination.

### 2.6. Antiproliferative Activity Assays

Antiproliferative experiments were performed on two different cell lines: human melanoma cells (A375, ATCC, CRL-1619™) and human normal dermal fibroblasts (HDFa, ATCC, PCS-201-012™). These cell lines were selected since melanoma has a high incidence in the population, and it is among the most severe skin cancers worldwide [28]. A375 and HDFa were grown in an RPMI (Roswell Park Memorial Institute) medium 1640 completed with 10% FBS (Fetal Bovine Serum). Penicillin (100 units/mL) and streptomycin (100 µg/mL) were added to the cell medium. Cells were grown in a 5% CO_2_ atmosphere at 37 °C using the CelCulture^®^ CO_2_ Incubator and allowed to reach a confluence of <70% in cell culture flasks with vented filter caps. Before treatments, adherent cells were detached using trypsin (1X), harvested in a 15 mL falcon tube, counted, and seeded in 96-well plates (8 × 10^3^ cells × well^−1^, with a final volume of 100 µL for each well). 96-well plates were incubated in a 5% CO_2_ atmosphere at 37 °C overnight. Total extracts of microalgal biomasses obtained from the Multicultivator growth conditions were dissolved in DMSO and used for all cell treatments in biological triplicate. The final concentration of the DMSO used was <0.5% (*v*/*v*) for each treatment. Cells were treated at 1, 10, and 100 µg/mL with total extracts (in technical triplicates for each biological replicate) for 48 h in a complete cell medium. Control cells were incubated in a complete cell medium with (positive control) and without (negative control) 0.5% of DMSO for all experiments.

#### Cell Viability

The antiproliferative effect of the samples on cell viability was evaluated using the 3-(4,5-Dimethylthiazol-2-yl)-2,5-diphenyl tetrazolium bromide (MTT) assay. Briefly, at the end of the incubation of the A375 and HDF with extracts, 10 µL of MTT solution (final concentration of 0.5 mg/mL) were added to each well. Plates were incubated in a 5% CO_2_ atmosphere for 3 h at 37 °C. After incubation, the MTT solution and media were removed using a vacuum aspirator system, and formazan salts produced by viable cells were dissolved in an isopropanol solution (100 µL) and incubated at room temperature for 30 min on an orbital shaker. The absorbance of each well was read at 570 nm using a ELX800 Absorbance Microplate Reader (Biotek, Winoosky, VT, USA). The antiproliferative effect of the extracts at different concentrations was reported as a percent of cell viability, calculated as the ratio between the mean absorbance of each treatment and the mean absorbance of the positive control (cells treated with only 0.5% of DMSO).

### 2.7. Liquid Chromatography—Mass Spectrometry Analysis

The extracts were analyzed using the Agilent high-performance liquid chromatography/mass spectrometry/electrospray ionization/quadrupole time-of-flight (HPLC/MS/ESI/Q-TOF) system. Analytical-grade water, acetonitrile, and formic acid were used for HPLC. The HPLC system was an Agilent 1260 Infinity. A reversed-phase Phenomenex Luna C18(2) column (250 mm × 10 mm, 5 μm particle size) with a Phenomenex C18 security guard column (4 mm × 3 mm) was used. The flow rate was 0.5 mL/min, and the column temperature was set to 40 °C. The eluents used were formic acid–water (0.1:99.9, *v*/*v*) for phase A and formic acid–acetonitrile (0.1:99.9, *v*/*v*) for phase B. The gradient used was as follows: 0–30 min with 100% A isocratic; 30–35 min with 100% B isocratic; 35–37 min with 100% B isocratic; 37–40 min with 100% A isocratic; 35–55 min washing and reconditioning the column with 5% B. The injection volume was 100 μL. The eluate was monitored using MS Total Ion Chromatogram (TIC). Mass spectra were obtained with an Agilent 6540 Ultra High Definition (UHD) Q-TOF spectrometer equipped with a dual-source Dual Agilent Jet Stream (AJS) ESI operating in positive mode. N_2_ was used as the desolvation gas at 300 °C with a flow rate of 8 L/min. The nebulizer was set to 45 psig. The sheath gas temperature was set to 400 °C with a flow rate of 12 L/min. A potential of 2.6 kV was applied to the capillary for negative ion mode and 3.5 kV for positive mode. The fragmentor was set to 175 V. Mass spectra were recorded in a range of 100–3200 *m*/*z*. Metabolites from bioactive biomass were identified using the Metabolomic Workbench database. Metabolomic data were normalized and analyzed using MetaboAnalyst 4.0 (https://www.metaboanalyst.ca/, accessed on 1 February 2023), highlighting qualitative and quantitative variations in metabolites between both treatments and strains [29]. A 1 mM solution of caffeine in methanol was added as an internal standard to normalize peak areas and minimize instrumental error. Caffeine was chosen because it is a metabolite absent in microalgae, ensuring no interference with the sample’s native metabolic profile.

### 2.8. Statistical Analysis

Statistical analyses were performed to evaluate biomass yields, biological assays, and metabolite differences under various growth conditions. The comparison of mixotrophic and phototrophic growth conditions was carried out using appropriate statistical tests in GraphPad Prism 9.3.1 software (GraphPad Software, San Diego, CA, USA). For biomass yield in flask experiments, a one-way analysis of variance (ANOVA) was applied to assess differences among the four growth conditions. For the Multicultivator experiments, an unpaired *t*-test with Welch correction was used to compare the two growth conditions. Finally, for biological assays and metabolite analysis, a two-way ANOVA was performed to analyze the interaction between variables across the two growth conditions.

Significance was determined based on *p*-values, with results considered statistically significant for *p* < 0.05.

## 3. Results

### 3.1. Effect of Four Different Cultivation Conditions on Growth Rates and Biomass Yields in Flask System

During the first days (0–3) of cultivation, *P. tricornutum* cells grew similarly under all four conditions. However, from day 3 onwards, cells in phototrophy and mixotrophy with the addition of bicarbonate (i.e., GM+BIC and GM+BIC+GLY, respectively) exhibited similar enhanced growth patterns compared to the other conditions, which continued until day 6. Notably, from day 6, the growth of cells in mixotrophy with the addition of only glycerol (GM+GLY) and in GM+BIC+GLY was significantly enhanced compared to the other cultivation conditions (Figure 1a). This indicates that the addition of glycerol promoted the growth of *P. tricornutum* cells starting from day 6.

Similarly, during the initial days of cultivation (0–3), *Chlorella* sp. cells grew similarly under all four conditions. From day 3, the growth in GM+BIC and GM+BIC+GLY was significantly enhanced compared to the other conditions (Figure 1b). By contrast, *Chlorella* sp. grew similarly in GM+GLY compared to phototrophic control (i.e., GM). This suggests that the addition of bicarbonate positively influenced the growth of *Chlorella* sp.

For *N. granulata*, during the initial days of cultivation (0–4), the growth of cells was similar in all the tested conditions. However, from day 4 onwards, the growth in GM+GLY, GM+BIC, and GM+BIC+GLY was significantly enhanced compared to the phototrophic control GM (Figure 1c). This indicates that both organic and inorganic carbon highly affected the growth of *N. granulata*.

In terms of biomass yield, GM+BIC and GM+BIC+GLY conditions consistently outperformed the others across all three model species (Figure 1d–f). For this reason, these conditions were upscaled in the Multicultivator system.

### 3.2. Effect of the Best Cultivation Conditions on Growth Rates and Biomass Yields in the Multicultivator System

Initial flask cultivation tests identified GM+BIC and GM+BIC+GLY as the most favorable conditions. Consequently, these conditions were scaled up using a Multi-Cultivator MC 1000 OD. Between days 4 and 5 of cultivation, significant growth enhancement was observed for *N. granulata *and *P. tricorntum* cells in GM+BIC+GLY compared to GM+BIC (Figure 2a,c). In contrast, *Chlorella* sp. cells exhibited similar growth under both conditions (Figure 2b).

Biomass yield analysis further confirmed that *N. granulata *and *P. tricornutum* demonstrated improved biomass yield in GM+BIC+GLY compared to GM+BIC (Figure 2d,f), aligning with previous results obtained in the Multi-Cultivator (Figure 2a,c). These findings suggest that both *N. granulata *and *P. tricornutum* can grow mixotrophically using glycerol as a carbon source. Conversely, *Chlorella* sp. appears unable to utilize glycerol for mixotrophic growth under the tested conditions.

The biomass dry weight and corresponding yields relative to the carbon concentration in the medium were analyzed in GM+BIC and in GM+BIC+GLY conditions. Table 1 summarizes the results, including the final pH values, highlighting species-specific differences in biomass yield per g of available carbon. Biomass yield is expressed as the percentage of dry weight (g/L) obtained per the concentration of available carbon (gC/L) in the medium. *P. tricornutum* exhibited the highest biomass conversion yield in GM+BIC condition (~82%), while the addition of glycerol in GM+BIC+GLY reduced the yield to ~22%, despite increased dry weight. Similarly, *N. granulata *achieved about 60% under GM+BIC, dropping to about 16% in GM+BIC+GLY. In contrast, *Chlorella* sp. demonstrated lower efficiency under GM+BIC (45.2%) and a striking drop to 9.7% in GM+BIC+GLY. This trend highlights strain-specific differences in metabolic pathways for utilizing inorganic and organic carbon sources. The pH of the medium plays a critical role in the availability of dissolved inorganic carbon, such as BIC and CO_2_. While pH was not actively controlled during cultivation, measurements at the beginning and end of the experiments revealed that all strains showed final pH values that ranged from 8.57 to 8.86. These conditions likely favored bicarbonate utilization, which is consistent with the relatively high yields observed in the GM+BIC condition. Indeed, in the pH range of 8 to 9, the predominant species is BIC, with minor contributions from carbonate (CO_3_^2−^), as carbonic acid (H_2_CO_3_) is almost negligible at this alkaline pH [30].

### 3.3. Antibacterial Activity

Total extracts from *N. granulata*, *Chlorella* sp., and *P. tricornutum* grown under GM+BIC and GM+BIC+GLY conditions were tested against two model pathogenic species: the Gram-positive *S. aureus* and the Gram-negative *E. coli*. Figure 3a–c illustrates the antibacterial activity of the total extracts against *S. aureus*. 2 mg/mL of the total extract from *P. tricornutum* under both GM+BIC and GM+BIC+GLY conditions reduced *S. aureus* cell viability by approximately 70% (Figure 3a). Similarly, the total extract from *Chlorella* sp. at a concentration of 2 mg/mL reduced *S. aureus* cell viability by approximately 40% under both conditions (Figure 3b). In contrast, the total extract from *N. granulata* under GM+BIC+GLY conditions exhibited only slight cytotoxicity against *S. aureus* (Figure 3c).

None of the total extracts from *N. granulata*, *Chlorella* sp., or *P. tricornutum* grown under GM+BIC or GM+BIC+GLY significantly affected the viability of *E. coli* (Figure 3d–f).

### 3.4. Antiproliferative Effects

Total extracts from *P. tricornutum*, *Chlorella* sp., and *N. granulata* grown under GM+BIC and GM+BIC+GLY conditions were tested on normal human dermal fibroblast cells (HDFa) and human melanoma cancer cells (A375).

Figure 4a–f presents the results of the viability assay after 48 h of treatment. Extracts from *P. tricornutum* exhibited mild antiproliferative effects on A375 cancer cells at all tested concentrations (100, 10, and 1 µg/mL). At the highest concentration, *P. tricornutum* reduced the viability of A375 cells by approximately 37 and 28% in GM+BIC and GM+BIC+GLY conditions, respectively. At lower concentrations, the cytotoxicity of *P. tricornutum* extracts decreased to around 10% in both conditions. Notably, the *Pt* extracts did not induce more than 20% mortality in normal cells under any treatment condition. At the highest concentration (100 µg/mL), *P. tricornutum* extracts caused a reduction in normal cell viability of about 20% in both GM+BIC and GM+BIC+GLY conditions. At medium (10 µg/mL) and low (1 µg/mL) concentrations, the reduction in normal cell viability was even lower (<12%, Figure 4d).

*Chlorella* sp. extracts exhibited a low antiproliferative effect on A375 cells under both GM+BIC and GM+BIC+GLY conditions (Figure 4b). However, these extracts showed strong cytotoxic effects on normal HDFa, particularly at high (100 µg/mL) and medium (10 µg/mL) concentrations. This cytotoxicity was especially pronounced with extracts from *Chlorella* sp. grown in GM+BIC+GLY (Figure 4e). *Chlorella* sp. extracts demonstrated greater cytotoxicity in normal cells than in cancer cells.

*N. granulata* extracts, grown under both phototrophic (GM+BIC) and mixotrophic (GM+BIC+GLY) conditions, did not produce metabolites with strong cytotoxic effects on A375 cancer cells. Only a slight reduction in A375 cell viability was observed at the highest concentration (100 µg/mL), with a decrease of 30% for GM+BIC and 25% for GM+BIC+GLY (Figure 4c). Similar effects were observed in normal HDFa treated with *N. granulata* extracts, where viability levels were comparable to those in cancer cells at 100 µg/mL. At medium and low concentrations (10 and 1 µg/mL), *N. granulata* extracts induced minimal cytotoxicity in normal cells. Overall, *N. granulata* extracts lacked a selective antiproliferative effect (Figure 4f).

### 3.5. Metabolite Analysis

The most bioactive extracts, specifically those of *P. tricornutum* and *Chlorella* sp. in GM+BIC and GM+BIC+GLY, were analyzed to identify their metabolite composition. A total of 44 metabolites were found in *P. tricornutum* from the analysis, as listed in Appendix A. The analysis identified a range of compounds, including mono- and digalactosyldiacylglycerides (e.g., MGDG (18:5/18:4) and DGDG (18:1/14:0)), fatty acids (e.g., PA (18:2/18:2)), pigments (e.g., fucoxanthin, pheophytin a), amino acids (e.g., leucine, phenylalanine), and other potential bioactive molecules. The metabolite analysis of *Chlorella* sp. grown under GM+BIC and GM+BIC+GLY conditions revealed a total of 34 metabolites. These included a diverse range of compounds, such as lipids, pigments, amino acids, depsipeptides, and other bioactive molecules. Several lipid classes were identified, including MGDGs and DGDGs, sphingolipids, and triacylglycerides (TAGs). Amino acids such as proline, leucine, and proline betaine and six classes of depsipeptides (I–VI) were also identified. Different pigments were identified, such as fucoxanthinol, violaxanthin, lutein, and the chlorophyll derivatives pheophytin a/a’ (Appendix A).

The graph indicates that the metabolic profiles of the total extracts from the two conditions, GM+BIC and GM+BIC+GLY, are similar, with only minor differences observed in a few cases. For example, in the total extract under mixotrophic conditions, pheophytin a’ is found in higher quantities compared to the total extract under phototrophic conditions. Conversely, the compound phosphatidic acid (PA) (18:2/18:2) shows the opposite trend (Figure 5a). This differential metabolic profile can be attributed to the mixotrophic condition (i.e., GM+BIC+GLY), which induces changes in both photosynthesis and lipid metabolism. In *Chlorella* sp., TAG (48:2) displayed a striking decrease in GM+BIC+GLY, suggesting significant changes in energy storage lipid dynamics. Another class of lipid, the dodecasphinganine, was significantly reduced in GM+BIC+GLY (Figure 5b).

## 4. Discussion

### 4.1. Biomass Yield and Carbon Source Utilization

This study represents the first comparative investigation of three industrial relevant marine microalgal strains—*P. tricornutum*, *Chlorella* sp., and *N. granulata*—under four distinct conditions: phototrophy (GM), mixotrophy (GM+GLY), phototrophy in presence of bicarbonate (GM+BIC) and mixotrophy in presence of bicarbonate (GM+BIC+GLY), focusing on both biomass yields and biological activity. While bicarbonate is widely studied as an inorganic carbon source to support both fresh and marine microalgae [26,31,32], this work provides novel insights into its combination with glycerol as an organic carbon source in mixotrophic system in these three selected species. Importantly, the study revealed strain-specific responses to these cultivation conditions, demonstrating the versatility of *P. tricornutum* and *N. granulata* in utilizing bicarbonate, while *Chlorella* sp. showed reduced carbon conversion efficiency (Table 1). Moreover, the combination of organic (GLY) and inorganic (BIC) carbon sources, alongside light, significantly optimized the growth of *P. tricornutum* and *N. granulata* (Figure 1a,c and Figure 2a,c). These results align with findings by [9], which demonstrated that combining glycerol and bicarbonate with enhanced micronutrient availability boosts both the quantity and quality of biomass in *P. tricornutum*. However, this study was limited to examining the effects of GLY and BIC on the growth and metabolites of *P. tricornutum*, without testing the differences in their biological activity. Similarly, *N. granulata* demonstrated increased biomass production under GM+BIC+GLY condition. Recent findings from [10] indicate that *N. granulata* observed similar growth rates under mixotrophic conditions, both in the presence and absence of CO_2_. The observed differences could be attributed to variations in experimental conditions, such as light intensity, medium composition, and concentration of inorganic carbon, which are known to significantly influence microalgal metabolism and growth [33]. For instance, even slight changes in light availability or nutrient composition could alter the balance of carbon source utilization, leading to variations in growth performance [34,35].

This improvement likely stems from the synergistic effect of utilizing dual carbon sources to fulfill energy and metabolic requirements, a strategy particularly beneficial for industrial applications. However, our results revealed distinct growth responses to different carbon sources in the tested conditions in *Chlorella* sp. Specifically, while bicarbonate alone improved growth for *Chlorella* sp., the combination of glycerol and bicarbonate did not yield a further enhancement (Figure 1b and Figure 2b). This suggests that under the experimental conditions tested, *Chlorella* sp. was unable to grow mixotrophically, implying a potential limitation in its ability to simultaneously utilize inorganic and organic carbon sources effectively. This observation is consistent with findings reported by [36], who demonstrated that supplying excess CO_2_ promoted the growth of *Auxenochlorella protothecoides* while simultaneously decreasing glycerol utilization. This supports the hypothesis that an abundance of inorganic carbon sources, such as bicarbonate or CO_2_, may inhibit or reduce the utilization of organic substrates in certain microalgal species. Such metabolic shifts could explain the growth patterns observed in our experiments and highlight the complexity of carbon source interactions in microalgal growth systems. However, a limitation of this study is the absence of real-time measurement of carbon source concentrations (i.e., GLY and BIC) during cultivation. While we inferred their utilization based on growth patterns and literature, future studies should include precise quantification of carbon source dynamics to validate the observed effects and enhance the mechanistic understanding of organic and inorganic carbon utilization. Additionally, the addition of glycerol in mixotrophic conditions led to reduced biomass yield obtained per the concentration of available carbon (g/gC, %), likely due to metabolic competition between the assimilation pathways of bicarbonate and glycerol. A previous study suggests that excess bicarbonate can decrease carbon utilization efficiency [31]. This is consistent with reports indicating that mixotrophic systems face trade-offs between substrate assimilation pathways. These trends align with findings from studies on *Microchloropsis gaditana*, where different cultivation modes (photoautotrophic, mixotrophic, and heterotrophic) significantly affected biomass productivity and substrate utilization [37].

### 4.2. Industrial Potential of Selected Microalgae for Nutraceutical Applications

Further analyses were performed to screen the possible biological activity of different microalgal extracts (Appendix A). Screening of extracts on cancer cell lines and their normal counterparts is essential as it could reveal a selective antiproliferative activity towards cancer cells, with no or slight cytotoxic effect on normal cells. Among the total extracts derived from the biomass of each of the three model species, the one from *P. tricornutum* showed both the highest antiproliferative effect on A375 cancer cells and the lowest cytotoxicity on human fibroblast cells (Figure 4a,d). Moreover, these extracts showed antibacterial activity against the pathogen *Staphylococcus aureus* ATCC25923.

This highlights *P. tricornutum*’s potential as a valuable source of bioactive compounds for pharmaceutical applications. Indeed, *P. tricornutum* is recognized as a valuable model organism for molecular research and synthetic biology applications due to its rich metabolic potential and adaptability to diverse growth conditions [38,39]. Various diatom species are a sustainable source of biologically active compounds with applications in pharmaceuticals, nutraceuticals, and cosmetics. They exhibit antioxidant, anti-inflammatory, anticancer, and antibacterial activities, including inhibiting melanoma cell proliferation and bacterial biofilm formation with minimal toxicity to human cells [40]. Unlike other diatoms, *P. tricornutum* does not exhibit sexual reproduction, which has significant implications for its technological potential and genetic manipulation [39,41,42]. However, although sexual reproduction has not been described for this species, it is possible that it occurs in some form, as the genome of this species contains meiosis-specific gene homologues [43]. The antimicrobial activity against *S. aureus* observed in this study may be linked to bioactive metabolites such as monogalactosyldiacylglycerol (MGDG) and corymbolone (Figure 3a and Figure 5a). MGDG has been previously reported to possess strong antibacterial activity against *S. aureus* [44], and its presence in *P. tricornutum* extracts supports this hypothesis. Moreover, a study on the cyanobacterium *Oscillatoria acuminata* demonstrated that MGDG-palmitoyl compromises bacterial membrane integrity, ultimately leading to cell lysis. This was confirmed through confocal laser scanning microscopy analysis, which revealed significant structural damage to the bacterial membrane of *Bacillus cereus* [45]. Additionally, corymbolone, a sesquiterpene keto-alcohol also identified in *P. tricornutum* extracts, has been shown to exhibit antibacterial properties against pathogenic strains [46].

The antiproliferative activity may be attributed to pheophytin, a chlorophyll derivative identified in the *P. tricornutum* extracts (Figure 4a and Figure 5a). Pheophytin has been reported in the literature as an effective antitumor compound, reducing the vitality of adenocarcinoma A549 cells while exhibiting minimal toxicity to normal cells [47]. This suggests that pheophytin could play a similar role in inhibiting A375 cell growth observed in this study, particularly in extracts from mixotrophic cultivation (Figure 4a). Chlorophylls and derivatives are known for their bioactive properties, including antioxidant, antimutagenic, and anticarcinogenic effects. Their distinct chemical structure enables them to neutralize harmful free radicals, reduce DNA damage, and regulate cellular mechanisms related to disease progression. Additionally, the hydrophobic side chains of chlorophylls promote interactions with biological membranes, affecting cellular uptake and signaling pathways [48,49].

The combination of these metabolites, along with other identified compounds, underscores the potential of *P. tricornutum* extracts as a source of antimicrobial and antitumoral agents.

While *P. tricornutum* extracts demonstrated the highest bioactivity, *Chlorella* sp. and *N. granulata* extracts revealed lower bioactivity. However, *Chlorella* sp. also showed antimicrobial activity against *S. aureus* (Figure 3b).

Moreover, *Chlorella* sp. extracts exhibited a low antiproliferative effect on cancer cells but notable cytotoxicity towards normal fibroblasts, particularly under mixotrophic conditions (Figure 4b,e). This suggests that *Chlorella* sp. metabolites may lack selectivity, posing challenges for therapeutic applications. This study revealed a high concentration of proline in *Chlorella* sp., confirming previous results [8]. Proline, a versatile amino acid, is known to play a role in osmoprotection, stress resistance, and energy metabolism in microorganisms. Its accumulation, under specific growth conditions, might enhance the organism’s adaptability to environmental stressors [50]. Additionally, proline derivatives like proline betaine may exhibit unique bioactivities, including antimicrobial properties, and could synergize with other compounds in the extract to inhibit microbial growth [51]. Moreover, proline-rich antimicrobial peptides are known for their antibacterial properties. These peptides can penetrate bacterial membranes and inhibit protein synthesis, leading to bacterial cell death. Their mode of action involves binding to bacterial ribosomes, thereby preventing the translation of essential proteins [52].

Additionally, Bibel et al. (1993) demonstrated that sphingosine inhibits *S. aureus* through lipid-mediated damage [53]. This mechanism involves the induction of multiple lesions in the bacterial cell wall, formation of membrane evaginations, and subsequent loss of ribosomes, further supporting the potential antimicrobial action of lipid-based metabolites found in the *Chlorella* sp. extract (Figure 5b).

*Nannochloropsis* species have garnered significant attention due to their high lipid content and industrial potential in biofuel production, aquaculture, and human nutrition. These microalgae are rich in essential fatty acids, proteins, and antioxidants, making them a valuable resource for functional foods and nutraceuticals [54]. Although *N. granulata* biomass increased significantly under mixotrophic conditions (Figure 2f), the extracts displayed limited antiproliferative activity on A375 cells and relatively low antibacterial activity (Figure 3c,f and Figure 4c). This suggests that while mixotrophic conditions boost biomass, the production of specific bioactive metabolites in *N. granulata* may require further optimization or additional stimuli (see summary in Table 2). Contrasting results were observed in [10], where mixotrophy enhanced the antitumoral activity of *N. granulata* on the PC3 prostate cell line. These differences may be attributed to variations in growth conditions and the specific target investigated.

## 5. Conclusions

In conclusion, this study highlights the significant potential of three marine microalgae as sources of bioactive compounds, with promising bioactivities observed in their methanol extracts. Among the tested species, *P. tricornutum* exhibited the most pronounced bioactivity, including antiproliferative and antibacterial effects against A375 and *S. aureus* cell lines, respectively, emphasizing its potential for pharmaceutical applications. Additionally, *Chlorella* sp. demonstrated notable bioactive properties, warranting further investigation. Optimizing cultivation conditions, such as refining carbon source combinations and environmental parameters, could enhance the production of bioactive metabolites in these species, thereby expanding their therapeutic and industrial applications. Future studies should expand the range of tested pathogenic strains, and conducting in-depth bioassays will further validate the antimicrobial and antiproliferative properties of the tested microalgae, enhancing their biotechnological applications in pharmaceuticals and nutraceuticals. Additionally, focus on the detailed chemical characterization and quantification of the bioactive compounds identified under mixotrophic conditions, exploring their mechanisms of action and potential synergistic effects.

## Figures and Tables

**Figure 1 microorganisms-13-00338-f001:**
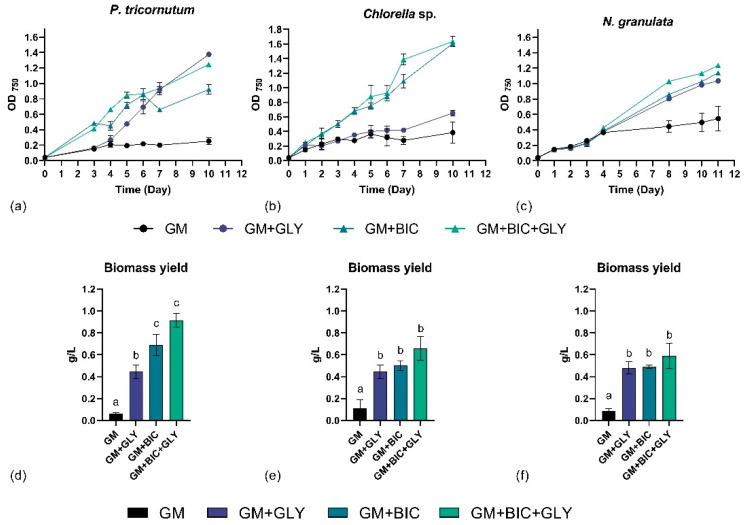
Growth profile of (**a**) *Phaeodactylum tricornutum* (*P. tricornutum*), (**b**) *Chlorella* sp., and (**c**) *Nannochloropsis granulata* (*N. granulata*); and biomass yield of (**d**) *Phaeodactylum tricornutum* (*P. tricornutum*), (**e**) *Chlorella* sp., and (**f**) *Nannochloropsis granulata* (*N. granulata*) in flask system comparing phototrophy (GM), mixotrophy with 4.6 g/L of glycerol (GM+GLY), phototrophy with 1.26 g/L of bicarbonate (GM+BIC), and mixotrophy with 1.26 g/L of bicarbonate and 9.21 g/L of glycerol (GM+BIC+GLY) cultivation conditions. Analyses were performed in biological triplicates, and the graphs represent means ± standard deviations. Different letters (a, b, c) denote significant differences among the two growth conditions (*t*-test, *p* < 0.05).

**Figure 2 microorganisms-13-00338-f002:**
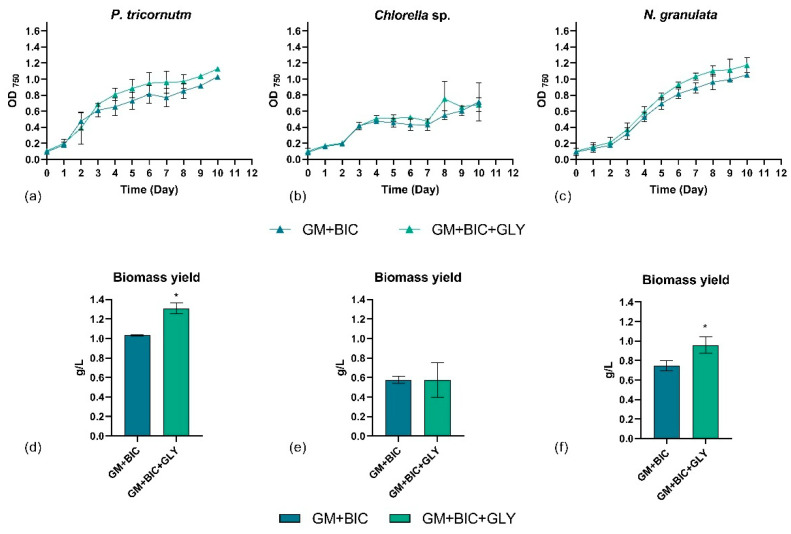
Growth profile of (**a**) *Phaeodactylum tricornutum* (*P. tricornutum*), (**b**) *Chlorella* sp., and (**c**) *Nannochloropsis granulata* (*N. granulata*), and biomass yield of (**d**) *Phaeodactylum tricornutum* (*P. tricornutum*), (**e**) *Chlorella* sp., and (**f**) *Nannochloropsis granulata* (*N. granulata*), in a Multicultivator system comparing phototrophy with 1.26 g/L of bicarbonate (GM+BIC) and mixotrophy with 1.26 g/L of bicarbonate and 4.6 g/L of glycerol (GM+BIC+GLY) cultivation conditions. Analyses were performed in biological triplicates, and the graphs represent means ± standard deviations. Differences between biomass yield in the two growth conditions were considered significant for *p*-values ≤ 0.05 (* *p* ≤ 0.05).

**Figure 3 microorganisms-13-00338-f003:**
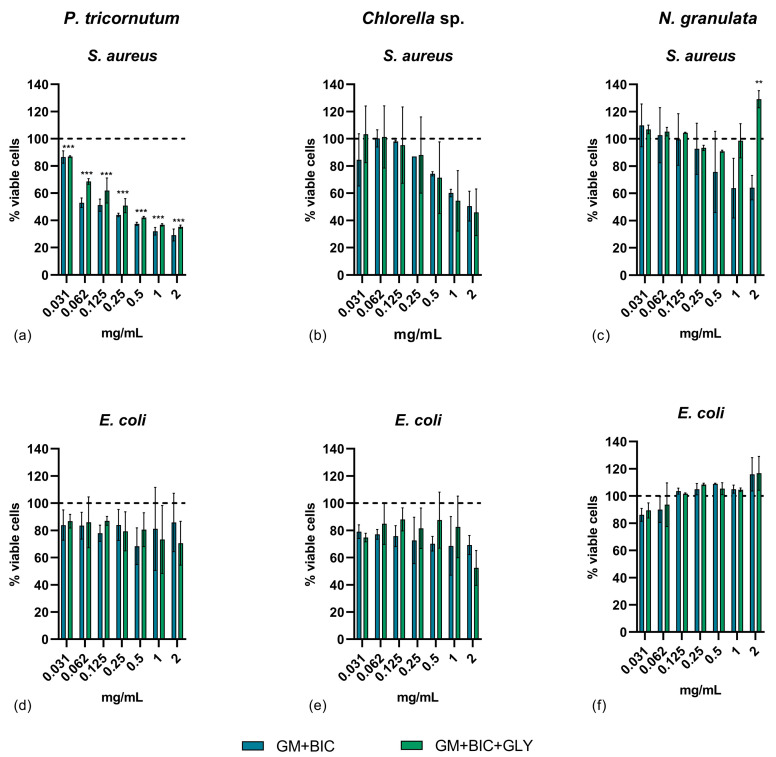
Antibacterial activity assay against *Staphylococcus aureus* ATCC25923 (*S. aures*) and *Escherichia coli* ATCC25922 (*E. coli*) cells after treatment for 24 h with 2, 1, 0.5, 0.25, 0.125, 0.0625, 0.03125, and 0 mg/mL (**a**,**d**) *Phaeodactylum tricornutum (P. tricornutum*), (**b**,**e**) *Chlorella* sp., and (**c**,**f**) *Nannochloropsis granulata (N. granulata)* extracts comparing phototrophy with 1.26 g/L of bicarbonate (GM+BIC) and mixotrophy with 1.26 g/L of bicarbonate and 4.6 g/L of glycerol (GM+BIC+GLY) cultivation conditions. Assays were performed in biological triplicates, and the graphs represent means ± standard. Dotted lines at 100% represent the negative control. Assays were performed in biological triplicates, and the graphs represent means ± standard deviations. Differences between the viability of *S. aureus* and *E. coli* were considered significant for *p*-values ≤ 0.05 (** *p* ≤ 0.01 and *** *p* ≤ 0.001).

**Figure 4 microorganisms-13-00338-f004:**
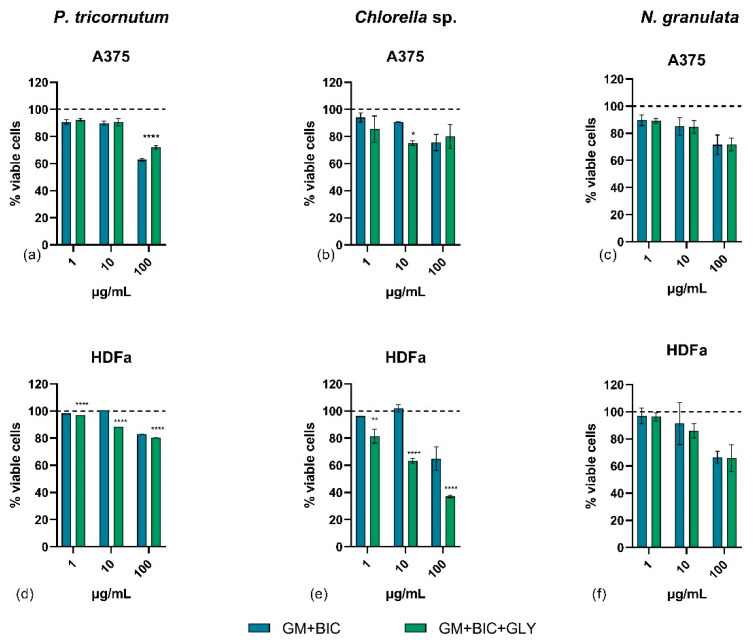
Cell viability assay on (**a**–**c**) A375 human melanoma cancer and (**d**–**f**) HDFa human dermal fibroblast cell lines. Bar graphs show the percentages of viable cells after 48 h of treatment with 1, 10, and 100 μg/mL of total extracts of Phaeodactylum tricornutum (*P. tricornutum*) (**a**,**d**), *Chlorella* sp. (**b**,**e**), and *Nannochloropsis granulata* (*N. granulata*) (**c**,**f**), comparing phototrophy with 1.26 g/L of bicarbonate (GM+BIC) and mixotrophy with 1.26 g/L of bicarbonate and 4.6 g/L of glycerol (GM+BIC+GLY) cultivation conditions. Cells treated with DMSO vehicle (0.5%) were used as a control and correspond to 100% of cell viability. Assays were performed in biological triplicates, and the graphs represent means ± standard deviations. Differences between the viability of A375 and HDFa were considered significant for *p*-values ≤ 0.05 (* *p* ≤ 0.05, ** *p* ≤ 0.01, and **** *p* ≤ 0.0001).

**Figure 5 microorganisms-13-00338-f005:**
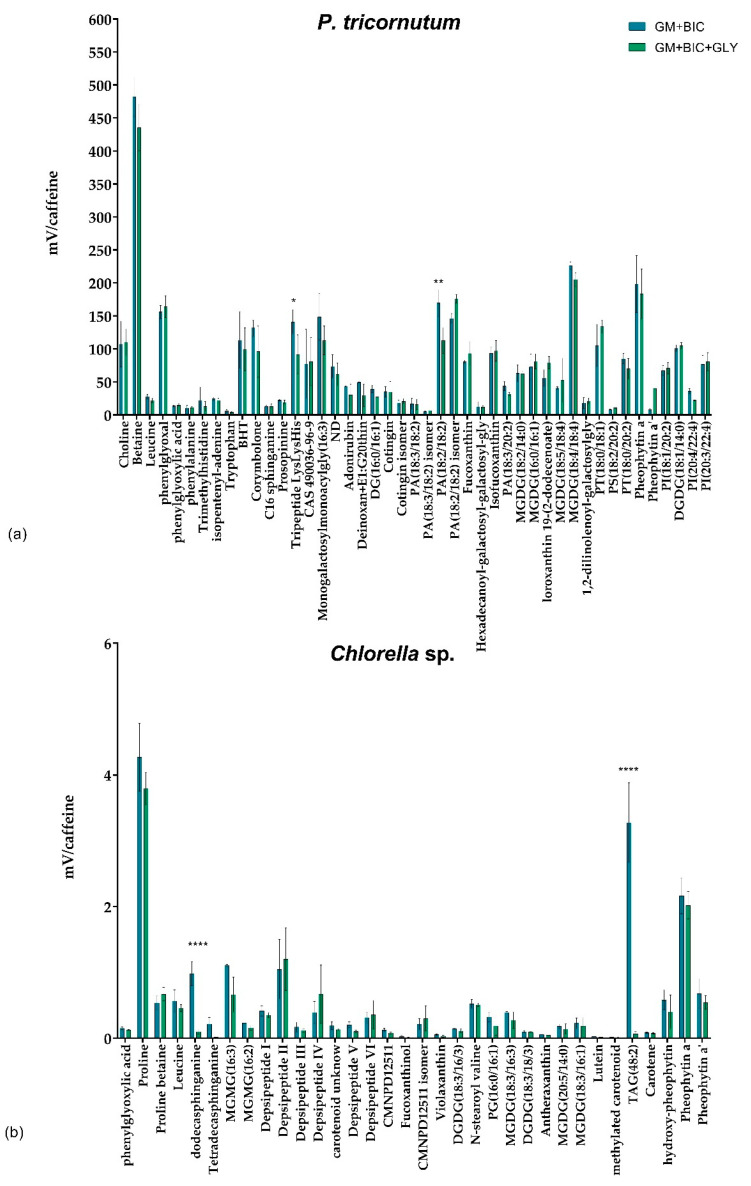
MS-HPLC analysis of (**a**) *Phaeodactylum tricornutum* (*P. tricornutum*) and (**b**) *Chlorella* sp. total extracts comparing phototrophy with 1.26 g/L of bicarbonate (GM+BIC) and mixotrophy with 1.26 g/L of bicarbonate and 4.6 g/L of glycerol (GM+BIC+GLY) cultivation conditions. Analysis was performed in biological triplicates, and the graphs represent means ± standard deviations. Differences between metabolite concentrations were considered significant for *p*-values ≤ 0.05 (* *p* ≤ 0.05, ** *p* ≤ 0.01, and **** *p* ≤ 0.0001). Data were normalized against a 1 mM caffeine solution in methanol to reduce instrumental error.

**Table 1 microorganisms-13-00338-t001:** Dry weight (g/L), concentration of available carbon (gC/L), biomass yield per g of available carbon (g/gC, %), and final pH of *Phaeodactylum tricornutum* (*P. tricornutum)*, *Chlorella* sp., and *Nannochloropsis granulata* (*N. granulata)* under phototrophic (GM+BIC) and mixotrophic (GM+BIC+GLY) conditions.

Species	Growth Condition	Dry Weight (g/L)	Carbon (gC/L)	Biomass Yield (g/gC, %)	Final pH
*P. tricornutum*	GM+BIC	1.03 ± 0.01	1.26	81.74% ± 0.80	8.86 ± 0.13
	GM+BIC+GLY	1.31 ± 0.05	5.86	22.35% ± 0.86	8.57 ± 0.05
*Chlorella* sp.	GM+BIC	0.57 ± 0.03	1.26	45.24% ± 2.38	8.84 ± 0.01
	GM+BIC+GLY	0.57 ± 0.18	5.86	9.73% ± 3.07	8.60 ± 0.28
*N. granulata*	GM+BIC	0.75 ± 0.05	1.26	59.52% ± 3.97	8.71 ± 0.20
	GM+BIC+GLY	0.96 ± 0.08	5.86	16.38% ± 1.37	8.60 ± 0.50

**Table 2 microorganisms-13-00338-t002:** Summary of growth conditions, biomass yield, antiproliferative activity, antibacterial activity, and key metabolites of the three studied microalgae species *Phaeodactylum tricornutum (P. tricornutum)*, *Chlorella* sp., and *Nannochloropsis granulata (N. granulata*). Biomass yield is presented as dry weight (g/L) under different carbon sources (bicarbonate (BIC) and bicarbonate + glycerol (BIC + GLY)). Antiproliferative activity is expressed as the percentage reduction in A375 cancer cell viability compared to normal fibroblast cytotoxicity. Antibacterial activity is reported as the percentage reduction in *Staphylococcus aureus* (*S. aureus*) viability. Key metabolites are compounds identified in extracts that may contribute to the observed bioactivities. Data highlights the species- and condition-specific differences in biomass production and bioactivity; monogalactosyldiacylglycerides (MGDG); n.a.: not available.

Species	Growth Condition	Biomass Yield	Antiproliferative Activity	Antibacterial Activity	Key Metabolites
*P. tricornutum*	BIC	Medium biomass yield (1.03 g/L ± 0.01)	Higher cytotoxicity against A375 cancer cells (up to ~40%) than on fibroblasts (up to ~20%)	Effective against antibacterial activity *S. aureus* (up to ~70%)	MGDG, fucoxanthin, pheothin a, corymbolone
	BIC + GLY	Significant increase in mixotrophy (1.31 g/L ± 0.05)	Higher cytotoxicity against A375 cancer cells (up to ~30%) than to fibroblasts (up to ~20%)	Effective antibacterial activity against *S. aureus* (up to ~65%)	MGDG, fucoxanthin, pheothin a, corymbolone
*Chlorella* sp.	BIC	Low biomass yield (<1 g/L)	Low cytotoxicity on A375 cells (up to ~25%) and on fibroblasts (up to ~35%)	Moderate antibacterial activity against *S. aureus* (up to ~50%)	Proline, pheothin a, sphingosine
	BIC + GLY	No significant enhancement in mixotrophy (<1 g/L)	Low cytotoxicity on A375 cells (up to ~20%) and high toxicity to fibroblasts (up to ~63%)	Moderate antibacterial activity against *S. aureus* (up to ~55%)	Proline, pheothin a, sphingosine
*N. granulata*	BIC	Low biomass yield (<1 g/L)	Low cytotoxicity on A375 cells (up to ~30%) and on fibroblasts (up to ~45%)	Low antibacterial activity against *S. aureus* (up to ~40%)	n.a.
	BIC + GLY	Significant biomass increase in mixotrophy (>1 g/L)	Low cytotoxicity on A375 cells (up to ~30%) and on fibroblasts (up to ~65%)	No antibacterial activity	n.a.

## Data Availability

The original contributions presented in this study are included in the article/Appendix A. Further inquiries can be directed to the corresponding author.

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
