# Peer review of "Mixotrophy in Marine Microalgae to Enhance Their Bioactivity"

_microorganisms, 2025, doi:10.3390/microorganisms13020338_

Round 1
Reviewer 1 Report
Comments and Suggestions for Authors
This paper presents an investigation into the potential of mixotrophic cultivation to enhance the bioactivity of marine microalgae species, specifically focusing on Phaeodactylum tricornutum, Chlorella sp., and Nannochloropsis granulata. The study demonstrates that mixotrophic conditions, combining light with both inorganic and organic carbon sources, significantly improve biomass production and metabolite diversity, leading to the discovery of bioactive compounds with antiproliferative and antibacterial activities. This research has important implications for the sustainable production of novel bioactive compounds for pharmaceutical and biotechnological applications.
The authors should consider the following concerns:
1. The paper lacks a graphical abstract, which is highly recommended to provide a concise overview of the study's objectives, methods, and main findings.
2. There are formatting issues throughout the text. For example, in line 91, the unit is not superscripted. Additionally, in line 446, "Csp" is not italicized consistently, as it is in some other parts of the text. The authors should carefully review the manuscript to ensure uniformity in formatting.
3. Lines 118-121 contain text that appears out of place or formatted incorrectly. The authors should review and correct this section.
4. The titling of the subheadings is inconsistent. Some subheadings (e.g., 2.1, 2.2, 2.3) do not have each word capitalized, while others (e.g., 2.4, 2.5) do. The capitalization of subheadings throughout the manuscript should be standardized.
5. The units of time (minutes, hours) are used inconsistently. Some instances use abbreviations (e.g., min, h), while others use the full word (e.g., minutes, hours). The authors should choose one format and use it consistently throughout the manuscript.
6. The choice of antimicrobial test strains (S. aureus and E. coli) is common but may not fully represent the diversity of potential pathogens. The authors should consider testing against a wider range of bacterial strains to verify the generalizability of their findings.
7. Metabolite analysis was limited to identifying compounds present in the extracts. The authors should consider performing a more in-depth analysis, such as quantifying the concentrations of bioactive compounds and exploring their synergistic effects.
8. The quality of the figures needs improvement. For example, figures 1 and 2 have misaligned panels and excessive whitespace between subfigures. Figures 3, 4, and 5 also contain excessive whitespace. The authors should optimize the figure layouts for clarity and aesthetics.
Additional concerns regarding the content and rigor of the study include:
9. The discussion of the results is limited and does not provide a thorough analysis of the mechanisms behind the observed bioactivities. The authors should delve deeper into the potential chemical and biological pathways leading to the antiproliferative and antimicrobial effects.
10. Mixotrophic culture conditions were limited to glycerol and bicarbonate as organic and inorganic carbon sources, respectively. The authors should explain the reasons for the choice of both and whether they are sufficiently representative.
Author Response
please find my answer in red in the attached document

Reviewer 2 Report
Comments and Suggestions for Authors
The study on " Mixotrophy in marine microalgae to enhance their bioactivity " is interesting, however, the manuscript can be improved by addressing the following observations:
Change the following keywords: marine microalgae; mixotrophy; bioactivity, because the keywords should not be repeated in the title.
Improve the quality of Figure 2 and 3.
In the conclusions section mention which would be the recommendations or bioassays for future research.
Author Response

(The authors gave the same response as above.)

Reviewer 3 Report
Comments and Suggestions for Authors
The manuscript presents a study on three marine algal species: chlorophytes, eustigmatophytes, and diatoms. It focuses on their growth in different trophic environments and the evaluation of the bioactivity of their extracts. This work is not only of fundamental importance, but also has great practical value. It contributes to the development of sustainable solutions in biotechnology, including the production of new antibacterial and anticancer drugs, which can have a significant impact on the health of people around the world. However, I have some comments that I would like to share. I hope these recommendations will help improve the manuscript and guide the authors in their future work.
General comments:
1. It remains a mystery to me why so many biotechnical studies do not include molecular identification of the strains they are studying. By skipping this crucial step, the authors cannot be completely certain they are referring to the right organisms. After all, errors can occur not only at the species level but also at the genus level, where alien properties may be mistakenly attributed to another taxon. DNA analysis has long been a cheap and quick procedure that should not be overlooked.
2. The abbreviations of the microalgae species' names, Phaeodactylum tricornutum (Pt), Chlorella sp. (Csp), and Nannochloropsis granulata (Ng), seem rather unfortunate to me. I find P. tricornutum, Chlorella sp., and N. granulata are much more familiar.
3. Please note that the link (https://aqualgae.com/es/productos-y-servicios/medios-de-cultivo-de-microalgas/) is not working.
4. For spectrophotometry, the optical path of the cuvette is usually specified.
5. What are the reasons behind choosing these specific forms of organic and inorganic carbon, and could you please explain this in your paper?
6. Also, it is not clear from the manuscript whether the algal cultures were axenic?
7. Could you please make the figures larger? They are currently too small.
8. Additionally, the origin of algae strains is not specified. Were these requested from a collection or were they personal isolates? And from which habitats were these isolates obtained?
9. Note the use of older nomenclature, such as Chlorella protothecoides instead of Auxenochlorella protothecoides. All species names should be checked against AlgaeBase (https://www.algaebase.org/).
10. I would recommend creating a summary table, perhaps even a heat map, of all three algae species, including their growth on different carbon sources and their antibacterial and antiproliferative effects.
11. Although the methods mention statistical analysis that was performed, the text does not further indicate whether the observed differences were statistically significant. The asterisks are only shown in Fig. 5, and have not been explained.
12. In my opinion, the manuscript doesn't discuss many key works on selected model organisms. For example,
Butler T, Kapoore RV, Vaidyanathan S. Phaeodactylum tricornutum: A Diatom Cell Factory. Trends Biotechnol. 2020;38(6):606-622. doi: 10.1016/j.tibtech.2019.12.023.
Russo MT, Rogato A, Jaubert M, Karas BJ, Falciatore A. Phaeodactylum tricornutum: An established model species for diatom molecular research and an emerging chassis for algal synthetic biology. J Phycol. 2023;59(6):1114-1122. doi: 10.1111/jpy.13400.
Lauritano C, Andersen JH, Hansen E, Albrigtsen M, Escalera L, Esposito F, Helland K, Hanssen KØ, Romano G and Ianora A (2016) Bioactivity Screening of Microalgae for Antioxidant, Anti-Inflammatory, Anticancer, Anti-Diabetes, and Antibacterial Activities. Front. Mar. Sci. 3:68. doi: 10.3389/fmars.2016.00068
Zanella L., Vianello F. Microalgae of the genus Nannochloropsis: Chemical composition and functional implications for human nutrition (2020) Journal of Functional Foods. 68: 103919, https://doi.org/10.1016/j.jff.2020.103919.
and many others.
13. What is also missing from the study is not only a comparison of the biotechnological potential of the strains under investigation with other species and genera, but also an analysis of their possible technological limitations. This includes, for example, the issue of sexual reproduction in diatoms.
14. Finally, I would like to draw attention to the carelessness in the manuscript's formatting. The authors should carefully check their text for compliance with the guidelines for authors and correct any errors, such as unnecessary use of italics, incorrect °C formatting, etc.
Specific comments:
1. L31: citotoxicity -> cytotoxicity
2. L52: intracelluar -> intracellular
3. L130, 140, 245, 269: in Multicultivator system -> in the Multicultivator system
4. L140: were re-suspended -> were resuspended
5. L148: died extracts -> dead extracts
6. L171: a confluence < 70% -> a confluence of < 70%
7. L173: harvested in 15 mL falcon tube -> harvested in a 15 mL falcon tube
8. L297: has high incidence in population -> has a high incidence in the population
9. L356: Another class of lipid, the dodecasphinganine was -> Another class of lipid, the dodecasphinganine, was
10. L393: normal cell -> normal cells
11. L404: sesquiterpenic -> sesquiterpene
12. L408: adenocarcinomic A549 cells -> adenocarcinoma A549 cells
Author Response

(The authors gave the same response as above.)

Reviewer 4 Report
Comments and Suggestions for Authors
References:
Sorry, but the list of references given at the end of the article contains a terrible design, so as it is impossible to adequately assess the correspondence of references to the topics discussed in the text with their mention.
There are no pages in 87% references (from 30): ##1,2,3,4,5,6,7,8,9,10,11,13,15,18,19,20,21,22,23,24,25,26,27,28,29,30.
Not only pages, but also the volume is absent in references ##18,22,15.
In case of reference #15 there is only name of the Journal, and all other information related to the publication (including year) is absent.
DOI is absent in 20% of the references (# 1,2,3,15,22,26).
The are 10 articles (## 6,7,8,9,10,14,18,20,23,24) among 30 references (30%) from the authors (please check it by the author Villanova). So, it is too much, and the amounts of such articles should be reduced in too times.
Of cause, all necessary information should be added to the article.
Introduction
There is no motivation for the choice of bicarbonate and glycerol (but not glucose, for example) for the cell growth. It shoud be added.
Methods
Line 91: The superscripts should be corrected in the phrase “20 μmol photons m−2 s−1.”
Lines 111-116: Please, add the values of pH for all variants of used media including the GM media. Generally, nothing is now about pH control in the work at all. That is bad.
Line 111: In the phrase “Phototrophy: Growth in GM without an external carbon source” nothing is about the type of external carbon source, concentration and its regulation/control (equipment, method?). How was the concentration of CO2 controlled in the flasks? Which stoppers were used in these flasks? All this information should be added.
Line 148: “dried” should be instead of “died”
Lines 155, 275, 278: All latin names of living organisms (Staphylococcus aureus, Escherichia coli) should be given in italic font.
Line 159: The superscript should be corrected in the phrase “4 × 105 CFU”.
Information about using of A375 human melanoma cancer and (d-f) HDFa human dermal fibroblasts cell lines for MTT should be added to the Methods.
Generally, all types of used equipment should be specified as well as its producer in the text.
Results
Lines 235-236: There is the phrase:“This suggests that the addition of bicarbonate positively influenced the growth of Csp.” This information is not already new, but it correlates with the published one (please see: “Kusi PA, McGee D, Tabraiz S, Ahmed A. Bicarbonate concentration influences carbon utilization rates and biochemical profiles of freshwater and marine microalgae. Biotechnol J. 2024,19(8), e2400361. doi: 10.1002/biot.202400361”). That should be mentioned in the text.
Talking about the biomass yield it is necessary to calculate it in % from the present carbon source! Of cause, if the summing concentration of carbon increases, so the biomass also accumulated better. However, the yield of biomass is interesting in relation to the used concentration of carbon! Please, calculate it and perform it in the article.
Figure 5: Please explain why the name of the ordinate axis is “mv/caffeine”? Why is caffeine present here?
The real results but not their treatment version for the MTT (like Figure 4) should be added to the Supplementary materials.
Discussion
I kindly ask the authors to definitely specify the novelty of results obtained! The authors compared these results with their own data obtained previously (references #9,24) and concluded that there is a difference (Lines 371-374). The reason of it is not definitely clear. Something influenced on the results obtained with the same cells in the same “working hands”. The reason should be disclosed deeper.
The big minuses of the work:
1- the authors discuss the influence of the conditions for cell cultivation and presence of the different concentrations of mixed carbon sources, but there is no control of these carbon sources (glycerol and CO2) in the used media at all! However, they use such information about carbon sources from other known publications in their Discussion section (Lines 383-389);
2- it is well known, that concentration of dissolved CO2 strongly depends on the pH value of the medium but the authors did not control it neither at the beginning, not during or at the end of cell cultivation.
Conclusion
Line 444: The information about the type of extract should be specified, because it is important and correct! It should be like this: “observed in their methanol extracts”.
Author Response

(The authors gave the same response as above.)

Reviewer 5 Report
Comments and Suggestions for Authors
Abstract
please comply with DMPI guidelines of 200 words
INTRODUCTION
idea on lines 35-37 requires citations
idea on lines 38-31 requires citations
METHODS
please be careful with the written of "μmol photons m−2 s−1."
please explain why that concentration of glycerol and bicarbonate were the only tested
how exactly was "scaled" the culture using the Multicultivator MC 1000 OD??
line 155. all scientific names must be written on italics
RESULTS
since figure 1 was created using PRISM, please add the One-way ANOVA analysis for figures D-E-F.
for figures 2. the correct analysis is not ANOVA, but T-TEST, in this case a unpaired T-TEST would be the best choice
for figure 3. change the graph to columns and perform the correct statistical analysis
for figure 4, perform Two-way ANOVA on PRISM
what about data of Nannochloropsis granulate on figure 5?
Author Response

(The authors gave the same response as above.)

Round 2
Reviewer 1 Report
Comments and Suggestions for Authors
None
Author Response
Thank you for taking the time to review and approve the revised manuscript.
Reviewer 3 Report
Comments and Suggestions for Authors
The authors fully considered my recommendations and I am pleased with the improvements made to the manuscript. Minor edits to the new version include the following: "sp." should not be italicized, and the abbreviations "Pt" and "Ng" have remained in some places in the text, namely at L303, 307, 325, 419 and 435.
In addition, I would like to emphasize that, although sexual reproduction has not been described for P. tricornutum, it is possible that it occurs in some form. At least, the genome of the strain CCAP 1055/1 contains genes responsible for meiosis, such as Mer3 helicase and Mnd1.
Author Response
The authors fully considered my recommendations and I am pleased with the improvements made to the manuscript. Minor edits to the new version include the following: "sp." should not be italicized, and the abbreviations "Pt" and "Ng" have remained in some places in the text, namely at L303, 307, 325, 419 and 435.
Thank you for your attention, we have now corrected all the suggested changes.
In addition, I would like to emphasize that, although sexual reproduction has not been described for P. tricornutum, it is possible that it occurs in some form. At least, the genome of the strain CCAP 1055/1 contains genes responsible for meiosis, such as Mer3 helicase and Mnd1.
Thank you for your comments, we have revised text in L531-533
Reviewer 4 Report
Comments and Suggestions for Authors
The authors made a notable work with the text of article but I still have some recommendations and comments for the authors:
- The following text appeared at Line 116: “mixing and aeration throughout the experiment.” Such actions were undertaken to provide the solubility of air in the media with the microalgae cells. So, by this way (through the “aeration”) the authors provided the phototrophic cells with carbon source such as CO2!? However, the information about the carbon concentration, its control or theoretical calculation is absent! The authors compared the influence of the carbon source and did not controlled this parameter… Moreover, the pH values of the used media with such “aeration” were different: from 6.7 to 8.3 (see the Lines 118-124). Since then, the obviously different soluble concentrations of CO2 were in the media in the experiments of the authors and they did not pay attention to this fact. Please, see the following publication (Figure 1): “Dasaard, C., Bayless, D., Stuart, B. (2016). Saturated pH and Total Inorganic Carbon from CO2 Solubility Related to Algal Growth. IARJSET, 3(11), 146-150. DOI:10.17148/IARJSET.2016.31128.” The conversion of CO2 to bicarbonate in relation to pH of medium GM is also well-known fact. Authors do not discuss that. The situation is very serious, because it is impossible to calculate the “𝐵𝑖𝑜𝑚𝑎𝑠𝑠 𝑦𝑖𝑒𝑙𝑑 per unit of available carbon” (Line 176).
Author Response
- The following text appeared at Line 116: “mixing and aeration throughout the experiment.” Such actions were undertaken to provide the solubility of air in the media with the microalgae cells. So, by this way (through the “aeration”) the authors provided the phototrophic cells with carbon source such as CO2!? However, the information about the carbon concentration, its control or theoretical calculation is absent!
The text in L116 referred to flask culture. The low mixing rate (100 rpm) in sealed culture flasks with vented caps was not intended to provide high CO2 concentrations. Instead, it was implemented to ensure uniform mixing and prevent sedimentation of microalgae cells. Based on this, atmospheric CO2 contributions were expected to be minimal, though not explicitly measured in this study.
The authors compared the influence of the carbon source and did not controlled this parameter… Moreover, the pH values of the used media with such “aeration” were different: from 6.7 to 8.3 (see the Lines 118-124). Since then, the obviously different soluble concentrations of CO2 were in the media in the experiments of the authors and they did not pay attention to this fact. Please, see the following publication (Figure 1): “Dasaard, C., Bayless, D., Stuart, B. (2016). Saturated pH and Total Inorganic Carbon from CO2 Solubility Related to Algal Growth. IARJSET, 3(11), 146-150. DOI:10.17148/IARJSET.2016.31128.”
The conversion of CO2 to bicarbonate in relation to pH of medium GM is also well-known fact. Authors do not discuss that. The situation is very serious, because it is impossible to calculate the “??????? ????? per unit of available carbon” (Line 176).
The significant differences in pH were mainly observed in the flask cultivation system, which lacked a precise aeration setup (L107-120). The biomass yield per unit of available carbon was calculated only in the multicultivation system, where GM+BIC and GM+BIC+GLY were tested, in these conditions pH remained stable during the cultivation time within the range of 8-9 (see table 1), this information was added in line L161. This range favors the bicarbonate form of carbon, as already discussed in the revised manuscript (lines 346-348). This is also in line what observed in the suggested article “Dasaard, C., Bayless, D., Stuart, B. (2016). Saturated pH and Total Inorganic Carbon from CO2 Solubility Related to Algal Growth. IARJSET, 3(11), 146-150. DOI:10.17148/IARJSET.2016.31128.” Furthermore, the composition of the GM medium was not provided in the suggested article, and acronym GM there refers to a general growth medium.